# Molecular Characteristics of *JAK2* and Its Effect on the Milk Fat and Casein Synthesis of Ovine Mammary Epithelial Cells

**DOI:** 10.3390/ijms25074027

**Published:** 2024-04-04

**Authors:** Yuan Liu, Huimin Zhen, Xinmiao Wu, Jiqing Wang, Yuzhu Luo, Jiang Hu, Xiu Liu, Shaobin Li, Mingna Li, Bingang Shi, Chunyan Ren, Yuanhua Gu, Zhiyun Hao

**Affiliations:** Gansu Key Laboratory of Herbivorous Animal Biotechnology, Faculty of Animal Science and Technology, Gansu Agricultural University, Lanzhou 730070, China; ly482098@163.com (Y.L.); zhenhm@st.gsau.edu.cn (H.Z.); wuxinmiao2020@163.com (X.W.); luoyz@gsau.edu.cn (Y.L.); huj@gsau.edu.cn (J.H.); liuxiu@gsau.edu.cn (X.L.); lisb@gsau.edu.cn (S.L.); limn@gsau.edu.cn (M.L.); shibg@gsau.edu.cn (B.S.); renyaya86@126.com (C.R.); guyh202209@163.com (Y.G.)

**Keywords:** *JAK2*, proliferation, milk synthesis, casein

## Abstract

In addition to its association with milk protein synthesis via the Janus kinase-signal transducer and activator of transcription (JAK-STAT) pathway, *JAK2* also affects milk fat synthesis. However, to date, there have been no reports on the effect of *JAK2* on ovine mammary epithelial cells (OMECs), which directly determine milk yield and milk contents. In this study, the coding sequence (CDS) region of ovine *JAK2* was cloned and identified and its tissue expression and localization in ovine mammary glands, as well as its effects on the viability, proliferation, and milk fat and casein levels of OMECs, were also investigated. The CDS region of ovine *JAK2*, 3399 bp in length, was cloned and its authenticity was validated by analyzing its sequence similarity with *JAK2* sequences from other animal species using a phylogenetic tree. *JAK2* was found to be expressed in six ovine tissues, with the highest expression being in the mammary gland. Over-expressed *JAK2* and three groups of *JAK2* interference sequences were successfully transfected into OMECs identified by immunofluorescence staining. When compared with the negative control (NC) group, the viability of OMECs was increased by 90.1% in the pcDNA3.1-*JAK2* group. The over-expression of *JAK2* also increased the number and ratio of EdU-labeled positive OMECs, as well as the expression levels of three cell proliferation marker genes. These findings show that *JAK2* promotes the viability and proliferation of OMECs. Meanwhile, the triglyceride content in the over-expressed *JAK2* group was 2.9-fold higher than the controls and the expression levels of four milk fat synthesis marker genes were also increased. These results indicate that *JAK2* promotes milk fat synthesis. Over-expressed *JAK2* significantly up-regulated the expression levels of casein alpha s2 (*CSN1S2*), casein beta (*CSN2*), and casein kappa (*CSN3*) but down-regulated casein alpha s1 (*CSN1S1*) expression. In contrast, small interfered *JAK2* had the opposite effect to *JAK2* over-expression on the viability, proliferation, and milk fat and milk protein synthesis of OMECs. In summary, these results demonstrate that *JAK2* promotes the viability, proliferation, and milk fat synthesis of OMECs in addition to regulating casein expression in these cells. This study contributes to a better comprehension of the role of *JAK2* in the lactation performance of sheep.

## 1. Introduction

Milk products mainly come from dairy cows, buffalos, dairy goats, sheep, and camels. While sheep milk holds a relatively small market share when compared to milk from dairy cows and goats, it has nearly double the fat and protein contents of the milk described above. Moreover, the size of the fat globules in sheep milk is only one-third of the size of those in cow milk, making its nutritional composition more similar to the breast milk of humans. Additionally, sheep milk contains a higher proportion of unsaturated fatty acids that are very beneficial for the elderly as they help prevent atherosclerosis and the formation of blood clots. Sheep milk is also favorable to emulsification, which facilitates its easier absorption by chyle in the small intestine. Given these traits, sheep milk is increasingly gaining popularity among consumers. According to the 2022 data from the Food and Agriculture Organization of the United Nations, global sheep milk yield has experienced consistent growth from 2012 to 2022, with an annual yield of nearly 11 million tons in 2022. Globally, people are increasingly recognizing the nutritional advantages of sheep milk.

Mammary epithelial cells (MECs), located in the parenchyma of the mammary gland, are responsible for the synthesis and secretion of milk. More specifically, in the later stages of pregnancy, MECs absorb nutrients from the blood to synthesize milk fat and milk proteins [1]. In this context, the quantity and secretory activity of MECs determine the yield and components of milk in mammals [2].

It is well known that the number of MECs and their activity are regulated by many functional genes and signaling pathways, including by Janus kinase 2 (*JAK2*) [3], signal transducer and activator of transcription 5a (*STAT5A*) [4], multiple endocrine neoplasia type 1 (*MEN1*) [5], AKT kinase/mammalian target of rapamycin kinase (AKT/mTOR) [6], and rapidly accelerated fibrosarcoma/MAP kinase-ERK kinase/Mitogen-Activated Protein Kinases (Raf/MEK/MAPK) [7]. The Janus kinase-signal transducer and activator of transcription (JAK-STAT) pathway is involved in the division, apoptosis, proliferation, and survival of MECs, as well as mammary gland development and lactation [8,9]. Paten et al. [10] showed that the JAK-STAT and PPAR pathways were differently regulated in ovine mammary gland tissues between late pregnancy and early lactation, indicating that these pathways play key roles in the functional development of the ovine mammary gland. As an important member of the JAK-STAT pathway, *JAK2* plays an important role in the proliferation of MECs and in the regulation of milk protein synthesis. In a study conducted on mice, the absence of *JAK2* led to a 95% reduction in MEC count [3]. In another study, interference with *JAK2* inhibited the proliferation and viability of breast cancer cells and promoted their apoptosis [11]. In bovine mammary epithelial cells (BMECs), over-expression of suppressor of cytokine signaling 3 (*SOCS3*) inhibited JAK2 phosphorylation, eventually reducing casein beta (CSN2) levels [12]. Meanwhile, fatty acids of 1.2 mM reduced JAK2 phosphorylation and the abundance of the JAK2 protein in BMECs, which, in turn, decreased the abundance of milk protein CSN2 [13], suggesting a positive correlation between *JAK2* and CSN2 content.

At present, research on the effect of *JAK2* on the fat and protein contents of milk has mainly been investigating dairy cows. These studies are typically conducted by investigating the roles of single-nucleotide polymorphisms (SNPs) in *JAK2*. Variations in *JAK2* have been found to be associated with milk yield, milk fat, and milk protein content. For example, milk yield, milk fat content, and milk protein content from cows of the *GG* genotype in c. 3232 A/G of *JAK2* were all higher than in cows of the *AG* and *AA* genotypes [14]. Moreover, Khan et al. [15] found that the variants in the *JAK2* of g. 39652267 C/T, g. 39645396 C/T, and g. 39631175 T/C significantly affected the milk fat percentage of Chinese Holstein cattle. However, no reports have been published on the direct influence of *JAK2* on milk protein content in sheep.

Given that the nucleotide sequences of ovine *JAK2* have only been predicted, its molecular function in ovine mammary epithelial cells (OMECs) remains unclear. Accordingly, in this study, we cloned the coding sequence (CDS) region of ovine *JAK2* and analyzed the molecular characteristics of the gene. More importantly, we investigated the effects of *JAK2* on the viability, proliferation, milk fat synthesis, and casein expression of OMECs.

## 2. Results

### 2.1. CDS Region Clone of Ovine JAK2 and Nucleotide Sequence Analysis

Using cDNA originating from ovine mammary glands as a template, a specific amplicon of 3635 bp was successfully cloned (Figure 1A). When compared to the predicted ovine *JAK2* mRNA sequences (GenBank accession no. XM_042243421.2), two SNPs (c.531 A/G and c.588 G/A) were identified in the cloned sequence obtained in this study (Figure 1B). As shown in Figure 1C, the sequences obtained in this study had similarities of 99.56%, 98.73%, 98.68%, 94.82%, 93.09%, 93.06%, 92.88%, 87.73%, and 81.08% with *JAK2* sequences from cattle, goats, yaks, pigs, humans, rhesus monkeys, chimpanzees, mice, and chickens. Phylogenetic analysis results further revealed that the sequences from this study shared a high similarity with the *JAK2* sequences from goats, cattle, and yaks (Figure 1C,D). These results validate that the ovine sequence obtained in this study represents the ovine *JAK2* sequence. The sequence has been submitted to GenBank with the accession number OR900882. Sequence analysis further revealed that the CDS region of ovine *JAK2* is 3399 bp in length.

### 2.2. Analysis of Predicted Ovine JAK2 Protein Characteristics

The cloned ovine *JAK2* sequences encoded polypeptides of 1132 amino acid residues in length. These contained high levels of leucine (9.63%), glutamic acid (7.86%), and lysine (7.16%). Tryptophan was the least common amino acid, with a frequency of 1.24% (Figure 2A). The JAK2 protein had a molecular formula of C_5872_H_9128_N_1586_O_1707_S_54_ and its isoelectric value and instability index were 6.84 and 45.01, respectively. The maximum hydrophobic value (2.311) was observed at position 288 while the minimum hydrophobic value (−3.111) was found at position 687 in the amino acid sequences. The number of hydrophobic amino acid residues was less than the hydrophilic amino acid residues, showing that JAK2 is a hydrophilic protein (Figure 2B). 

### 2.3. Localization of JAK2 in the Ovine Mammary Gland and Its Expression in Tissue

The immunofluorescence result revealed that the ovine JAK2 protein was mainly distributed in OMECs surrounding the glandular cavity (Figure 3). The RT–qPCR results illustrated that *JAK2* exhibited a tissue-specific expression pattern in sheep. *JAK2* was not expressed in the lungs. Of the remaining tissues, *JAK2* had the highest expression level in the mammary gland and the lowest in the ovaries (*p* < 0.05, Figure 4).

### 2.4. Culture and Identification of OMECs

Fibroblasts were removed via 0.25% trypsin digestion at 3, 5, and 7 days after culture, resulting in OMECs with a cobblestone-like morphology (Figure 5A). The results showed that CK18 was positively expressed in the cytoplasmic region of the cultured cells (Figure 5B). These findings show that pure OMECs were successfully cultured.

### 2.5. JAK2 Increases the Viability of OMECs

After transfecting for 48 h, the expression of *JAK2* was 191-fold higher in the pcDNA3.1-*JAK2* group compared with the pcDNA3.1-NC group (*p* < 0.01, Figure 6A). In contrast, the expression of *JAK2* decreased in three groups of small interfered *JAK2* (*p* < 0.01, Figure 6B), among which si-*JAK*2-1 was the most prominent and, therefore, selected for subsequent interference experiments. The findings show that over-expressed and interfered *JAK2* were all successfully transfected into OMECs. A CCK-8 assay showed that the viability of OMECs was increased by 90.1% in the pcDNA3.1-*JAK2* group compared with its NC group (*p* < 0.01, Figure 6C) while the viability was decreased by 32.7% in the si-*JAK*2-1 group (*p* < 0.01, Figure 6C).

### 2.6. JAK2 Promotes the Proliferation of OMECs

The results of the EdU assay revealed that the over-expression of *JAK2* increased the number (Figure 7A) and ratio (*p* < 0.01; Figure 7B) of EdU-labeled positive OMECs compared to the NC group. Interference with *JAK2* yielded the opposite effect. Meanwhile, interfered *JAK2* resulted in significant decreases in the expression levels of the cell proliferation marker genes cyclin-dependent kinase 2 (*CDK2*) (*p* < 0.05), cyclin-dependent kinase 4 (*CDK4*) (*p* < 0.01), and proliferating cell nuclear antigen (*PCNA*) (*p* < 0.01; Figure 7C). Conversely, over-expression of *JAK2* increased the expression levels of these genes (*p* < 0.01; Figure 7C). These results confirm that *JAK2* promotes the proliferation of OMECs.

### 2.7. Effect of JAK2 on Milk Fat Synthesis

To evaluate the effect of *JAK2* on milk fat synthesis, the triglyceride contents and expression levels of marker genes related to milk fat synthesis were investigated. The results showed that the triglyceride content in the over-expressed *JAK2* group was 2.9-fold higher compared to the NC group (*p* < 0.01; Figure 8A). Meanwhile, the expression levels of milk fat synthesis marker genes, namely, fatty acid synthase (*FASN*), peroxisome proliferator-activated receptor gamma (*PPARG*), stearoyl-CoA desaturase (*SCD*), and lipoprotein lipase (*LPL*), were increased by 2.1-fold (*p* < 0.01), 1.2-fold (*p* < 0.01), 1.1-fold (*p* < 0.01), and 1.5-fold (*p* < 0.05; Figure 8B), respectively, when pcDNA3.1-*JAK2* was transfected into OMECs. On the contrary, interfered *JAK2* decreased the triglyceride levels and the expression levels of the marker genes in OMECs. These findings reveal that *JAK2* promotes milk fat synthesis in OMECs.

### 2.8. JAK2 Regulates the Expression of Casein Genes

The over-expression of *JAK2* significantly up-regulated the expression levels of casein alpha s2 (*CSN1S*2), *CSN2*, and casein kappa (*CSN3*) and down-regulated casein alpha s1 (*CSN1S1*). The inhibition of *JAK2* had the opposite effect on the expression of these genes (*p* < 0.01; Figure 9). These findings show that *JAK2* has a regulatory effect on the expression of casein genes.

## 3. Discussion

In this study, we cloned the CDS region of ovine *JAK2* to determine its molecular characteristics, localization in the ovine mammary gland, and tissue expression, as well as its effects on the viability, proliferation, milk fat synthesis, and casein contents of OMECs. The cloned sequence encoded polypeptides of 1132 amino acids in length and shared the highest homology with the *JAK2* sequences originating from goats, cattle, and yaks. Based on these findings, it was confirmed that the cloned CDS region indeed represents the ovine *JAK2* sequence. Compared with the predicted ovine *JAK2* sequence (XM_042243421.2), there were two SNPs (c.531 A/G and c.588 G/A) in the sequences cloned in this study; this difference in nucleotide sequences may be related to sheep breed.

The analysis of amino acid composition showed that the ovine JAK2 protein contained a high proportion of leucine, glutamic acid, and lysine. Glutamic acid and lysine are strongly hydrophilic amino acids, which aligns with the prediction that the JAK2 protein is a hydrophilic protein. It has been reported that lysine is the main limiting amino acid for milk protein synthesis in dairy cows [16] and it was found that adding 1 mmol/L of lysine to the BMEC culture medium can improve milk protein synthesis [17]. Leucine promotes *CSN2* and *CSN1S1* synthesis at concentrations below 0.75 mmol/L [18]. These findings suggest that the higher levels of leucine and lysine in ovine JAK2 are closely related to its role in milk protein synthesis.

The immunofluorescence results from mammary gland tissues confirmed that JAK2 is mainly located in the lumina of MECs, where the components of milk are synthesized. Given that the quantity and activity of MECs are responsible for the yield and quality of milk [19], it is concluded that *JAK2* may play a role in the milk synthesis of sheep by influencing these factors.

The RT–qPCR results revealed that *JAK2* exhibits a tissue-specific expression pattern in sheep as the highest expression level of this gene was observed in mammary glands and the lowest in the lungs. The difference in the expression of *JAK2* across different tissues may be related to its biological function. The highest expression of *JAK2* being observed in the mammary glands suggests that the gene plays an important function in mammary gland development and lactation. This hypothesis was subsequently confirmed by investigating the effect of *JAK2* on the activity levels and milk fat and casein contents of OMECs. It was notable that *JAK2* also had a higher expression in the liver. This was not surprising as the mammary gland absorbs glucose synthesized by the liver and then converts it into lactose and glycerol, which are the backbones of milk fat [20]. In addition, the liver is the center of lipid metabolism, which not only oxidizes free fatty acids for energy [21] but also transports triglycerides to the mammary glands for use in milk fat synthesis. To date, no reports have been published on the expression levels of *JAK2* in the tissues of other species.

It is well known that the number and activity levels of MECs are positively correlated with milk yield [22]. Meanwhile, the proliferation and apoptosis of MECs are responsible for changes in the number of MECs during lactation [2]. In this context, the effect of *JAK2* on the viability and proliferation of OMECs attracted our attention. The CCK-8 assay showed that the over-expressed *JAK2* increased the viability of OMECs. The EdU results revealed that the over-expression of *JAK2* increased the number and ratio of EdU-labeled positive OMECs. Meanwhile, the over-expression of *JAK2* also increased the expression levels of *CDK2*, *CDK4*, and *PCNA*. These results suggest that *JAK2* promotes the proliferation of OMECs. Taken together with the roles OMEC activity and proliferation play in determining milk yield, it was speculated that *JAK2* may enhance milk yield in sheep. The *CDK2*, *CDK4*, and *PCNA* genes are usually used as cell proliferation marker genes when investigating the effects of RNAs on cell proliferation. For example, these three genes have been used to study the effect of cyclin-dependent kinase inhibitor 1B (*p27*) [23] and circ_015343 [24] on the proliferation of ovine preadipocytes and OMECs, respectively. Up to now, to our knowledge, research on the influence of *JAK2* on cell proliferation has mainly focused on human cancer cells. For instance, interfered *JAK2* was found to inhibit the proliferation of liver cancer cells, eventually reducing tumor size and weight [25]. Li et al. [26] found that the knockout of *JAK2* reduced the proliferation of gastric cancer cells. A lack of *JAK2* was found to inhibit the proliferation of mouse MECs by suppressing the accumulation of cyclin D1 protein [27]. The mechanism underlying this process may be that JAK2 initiates the transcription of AKT1 mRNA by activating STAT5. The activated AKT1 increases the expression levels of cyclin D1 by inhibiting the activity of glycogen synthase kinase 3β (GSK3β) and, thus, promoting cell proliferation [28].

The synthesis of mammalian milk proteins is primarily regulated by the JAK-STAT signaling pathway [29]. When prolactin, insulin, and other extracellular signals bind to their respective receptors on the cell membrane, JAK2 is phosphorylated and catalyzed, leading to the activation of the downstream protein STAT5 [30]. Activated STAT5 forms a dimer that translocates into the nucleus where it binds to specific sites in the promoter region of the target genes, initiating the transcriptional processes of various milk protein genes [31,32]. In this study, over-expression of *JAK2* resulted in increased expression levels of *CSN1S2*, *CSN2*, and *CSN3* but decreased expression levels of *CSN1S1*. Previous studies have shown that STAT5A, activated by the JAK2 protein, activates the transcription of *CSN2* by binding to its promoter region, thus promoting the synthesis of *CSN2* [33]. Meanwhile, another study showed that *CSN1S1* reduced the phosphorylation levels of JAK2 and STAT5A, thereby reducing the abundance of the CSN2 protein [34]. Meanwhile, regulating the expression and phosphorylation of key factors in the JAK-STAT pathway resulted in changes in intracellular milk protein contents. For instance, over-expression of *SOCS1* suppressor resulted in the down-regulation of the mRNA and protein abundances of *JAK2*, eventually decreasing the expression levels of *CSN2* and *CSN3* [35]. Huang et al. [36] further found that *SOCS3* suppressed *CSN2* expression by inhibiting the JAK2-STAT5A pathway. As a positive regulator, Met was shown to promote the synthesis of *CSN2* by activating the JAK-STAT signaling pathway [13]. This may explain why the expression of *JAK2* was positively correlated with *CSN2* and *CSN3* but negatively correlated with *CSN1S1*. However, there are exceptions. Bionaz et al. [29], for example, found the JAK2-STAT5 pathway had a relatively weak effect on milk protein synthesis in bovine mammary glands. These findings suggest that the effect of JAK2 on protein levels in milk needs to be investigated further.

Although *JAK2* plays a crucial role in milk protein synthesis, it was also found to be related to milk fat. Our results demonstrate that *JAK2*-1 increased the expression levels of *FASN*, *SCD*, *PPARG*, and *LPL.* Meanwhile, *JAK2*-1 was found to increase triglyceride contents, which make up 98% of milk fat. These findings suggest that *JAK2* promotes milk fat synthesis in OMECs. The expression levels of *FASN*, *SCD*, *PPARG*, and *LPL* are all positively associated with milk fat synthesis. For example, the FASN enzyme is a major source of short- and medium-chain fatty acids found in milk and is involved in the de novo synthesis pathway of the fatty acid metabolism of the mammary glands [37]. A decrease in *SCD* expression levels in the mammary gland was accompanied by decreased milk fat content [38,39]. Moreover, *PPARG* regulates energy metabolism and fat formation and stimulates the monounsaturated fatty acid synthesis pathway in the mammary glands [40,41,42]. The synthesized fatty acids are taken up by LPL from plasma. Meanwhile, LPL, in coordination with very low-density lipoprotein receptors (VLDLRs), enhances the activity of long-chain fatty acids [43].

## 4. Materials and Methods

### 4.1. Sample Collection

The Animal Experiment Ethics Committee of Gansu Agricultural University approved our experimental animal studies under approval number GSAU-ETH-AST-2021-027.

Three healthy two-year-old Hu ewes were selected, kept under the same feeding and management conditions, and were all in their second parity. During early lactation, the ewes were slaughtered to collect heart, liver, spleen, lung, kidney, ovary, and mammary gland tissues. The collected tissues were immediately preserved in liquid nitrogen for subsequent RNA extraction. At the same time, a part of the mammary gland parenchyma was collected to culture OMECs.

### 4.2. Molecular Clone and Characteristics Analysis of Ovine JAK2

Based on the predicted ovine *JAK2* sequences (GenBank accession no. XM_042243421.2) available in GenBank, two PCR primers (5′-GTTTCGGAAGCAGGCAAGG-3′ and 5′- TTTCCACACTTTTGCTGGCT-3′) were designed to amplify a specific fragment of 3635 bp that would contain a 3399 bp CDS region of the ovine *JAK2*. The cDNA originating from ovine mammary gland tissue was used to perform a PCR reaction with 2 × Rapid Taq Master Mix (Vazyme, Nanjing, China). Subsequently, PCR amplicons were detected using electrophoresis in 1.5% agarose gel (Tsingke, Beijing, China) and then purified using the gel extraction kit (Servicebio, Wuhan, China). The purified PCR products were cloned into pMD19-T vectors using the pMD^TM^19-T vector cloning kit (TaKaRa, Beijing, China). The products were sequenced by Yangling Biotech Co., LTD. (Yangling, China) in order to validate their authenticity.

The open reading frame (ORF) of the cloned ovine *JAK2* was identified using ORF Finder (NCBI, Bethesda, MD, USA). A homology comparison of the nucleotide sequences was performed using DNAMAN 8.0 (Lynnon Biosoft, San Ramon, CA, USA). A neighbor-joining phylogenetic tree was constructed based on the cloned sequence obtained in this study and *JAK2* sequences from nine other species using MEGA 7.0 software (Institute for Genomics and Evolutionary Medicine, Temple University, Philadelphia, PA, USA). The ovine *JAK2* CDS sequences were translated into amino acid sequences using the online ExPASy-Translate tool (Swiss Institute of Bioinformatics, Basel, Switzerland). The physical and chemical properties of the protein were analyzed using ExPASy-ProParam (Swiss Institute of Bioinformatics, Basel, Switzerland) while the hydrophilicity/hydrophobicity of the protein was investigated using ExPASy-Proscale (Swiss Institute of Bioinformatics, Basel, Switzerland).

### 4.3. RNA Extraction and Reverse Transcription–Quantitative PCR

The total RNA was extracted from seven ovine tissues and OMECs using TRIzol reagents (Invitrogen, Carlsbad, CA, USA). The concentration and purity of RNA were measured using a NanoDrop 8000 spectrophotometer (NanoDrop Technologies, Wilmington, NC, USA). Subsequently, cDNA was produced using SweScript All-in-One RT SuperMix for qPCR (Servicebio, Wuhan, China). The reverse transcription–quantitative PCR (RT–qPCR) analysis was conducted in triplicate with the 2 × SYBR Green qPCR Master Mix (Servicebio, Wuhan, China) on an Applied Biosystems QuantStudio 6 Flex Real-time PCR system (Thermo Fisher Scientific, Waltham, MA, USA). Additionally, *β*-actin was chosen as an internal reference gene. The relative expression levels of each gene were determined using the 2^−ΔΔCt^ method. The primer information for RT–qPCR is listed in Table 1.

### 4.4. Localization of JAK2 in Ovine Mammary Glands

The localization of JAK2 in ovine mammary glands was detected using immunofluorescence staining, as described by Li et al. [44]. Briefly, slices of ovine mammary gland tissue were dewaxed using a dewaxing solution (Servicebio, Wuhan China) and then treated with antigen retrieval. Subsequently, the slices were washed with phosphate-buffered saline (PBS, Servicebio, Wuhan, China) and air-dried before being blocked with 5% BSA (Servicebio, Wuhan, China). After 30 min, the tissues were incubated with rabbit primary antibody against JAK2 (1: 100; Abmart, Shanghai, China) at 4 °C overnight. Cy3-labeled goat anti-Rabbit IgG (1: 300; Servicebio, Wuhan, China) was added and then incubated at room temperature for 1 h. The slices were washed three times with PBS before being stained with DAPI solution (Servicebio, Wuhan, China) and, then, incubated at room temperature for 10 min. Finally, the slides were sealed with anti-fluorescence quenching tablets (Servicebio, Wuhan, China) and, then, observed using a fluorescence microscope (Olympus, Tokyo, Japan). 

### 4.5. Isolation, Culture, and Identification of OMECs

Ovine mammary gland tissues were washed three times with PBS and then cut into approximately 1 mm^3^ fragments using scissors. Trimmed tissues were placed in PBS containing type I collagenase (Solarbio, Beijing, China). The mixtures were incubated in an incubator (Zhichu, Shanghai, China) with a rotating speed of 200 rpm for 1 h. Subsequently, the mixture was filtered through 100 um and then 40 um cell strainers (Biosharp, Beijing, China). The filtrate was centrifuged and the pellet was resuspended in a complete culture medium containing fetal bovine serum (Invigentech, Carlsbad, CA, USA) and DMEM/F12 medium (Hyclone, Logan, UT, USA). The cells were then seeded into culture flasks (Servicebio, Wuhan, China) for further cultivation. Fibroblasts were removed by following the method described by Hao et al. [45].

The purity of the cultured cells was confirmed via immunofluorescence staining. In brief, the cells in the 12-well plate (Servicebio, Wuhan, China) were fixed with 4% paraformaldehyde for 20 min at room temperature. Subsequently, cells were permeabilized with 0.5% Triton X-100 (Servicebio, Wuhan, China) for 20 min and then blocked with 5% BSA (Solarbio, Beijing, China) at room temperature for 30 min. The cells were incubated with the rabbit anti-CK18 primary antibody (1: 100; Abmart, Shanghai, China) at 4 °C and then treated with goat anti-rabbit IgG FITC (1: 500; Abmart, Shanghai, China). After being incubated at 37 °C for 2 h, the cell nucleus was stained for 20 min using Hoechst 33342 dye (Beyotime, Shanghai, China) and then observed using an IX73 inverted fluorescence microscope (Olympus, Tokyo, Japan).

### 4.6. Construction of Over-Expressed and Small Interfered JAK2 Vectors 

The over-expressed vector of *JAK2* (named pcDNA3.1-*JAK2*) and its negative control (NC; named pcDNA3.1-NC) were synthesized by Suzhou Hongxun Biotechnology Co., LTD (Suzhou, China). The vector had NheI and BamHI restriction sites. Three *JAK2* interference sequences (named si-*JAK2*-1, si-*JAK2*-2, and si-*JAK2*-3) and their NCs (named si-NC) were also synthesized. The specific interference sequences can be found in Table 2.

### 4.7. Cell Transfection, EdU, and CCK-8 Assays

When the confluence of OMECs reached 70–80%, they were digested with 0.25% trypsin (Gibco, Carlsbad, CA, USA) and then centrifuged at 1200 rpm for 5 min. The precipitates were resuspended in a complete medium and uniformly seeded onto a cell culture plate. The pcDNA3.1-*JAK2*, si-*JAK2*-1, and their NC were, respectively, transfected into OMECs using INVI DNA RNA Transfection Reagent^TM^ (Invigentech, Carlsbad, CA, USA).

After 48 h of transfection, an EdU assay was performed using the BeyoClick^TM^ EdU-555 cell proliferation assay kit (Beyotime, Shanghai, China). The number and percentage of EdU-labeled positive OMECs were observed and calculated using an IX73 microscope (Olympus, Tokyo, Japan) and ImageJ 1.54d software (National Institutes of Health, Bethesda, Montgomery, MD, USA), respectively. Meanwhile, transfected OMECs with pcDNA3.1-*JAK2*, si-*JAK2*-1, and their NCs were supplemented with 10 μL of CCK-8 reagent (Vazyme, Nanjing, China) and further cultured in an incubator at 37 °C for 2 h. The absorbance of the OMCEs at 450 nm was measured using a microplate reader (Thermo Scientific, Waltham, MA, USA).

### 4.8. Triglyceride Content Detection

After transfecting pcDNA3.1-*JAK2*, si-*JAK2*-1, and their NCs into OMECs for 48 h, the supernatants of the OMECs were collected by disrupting the transfection on ice and centrifuging at 8000 g for 10 min. Subsequently, a cell/tissue/serum triglyceride assay kit (Solarbio, Beijing, China) and a microplate reader (Thermo Scientific, Waltham, MA, USA) were used to detect the triglyceride content in the OMECs.

### 4.9. Statistical Analysis

SPSS v24.0 (IBM, Armonk, NY, USA) and GraphPad Prism 8.02 (GraphPad Software, La Jolla, CA, USA) were utilized for data analysis and image generation, respectively. Two-tailed independent t-tests were used for comparisons between the two groups while one-way ANOVA was used for comparisons between multiple groups. The results are presented as mean ± standard errors of the mean (SEM).

## 5. Conclusions

*JAK2* had different expressions in different ovine tissues, with the highest level of expression found in mammary glands. In vitro cell experiments indicated that *JAK2* promoted the viability, proliferation, and milk fat synthesis of OMECs. The gene also influenced the expression levels of casein genes. This study provides a comprehensive understanding of the molecular mechanism of *JAK2* in ovine mammary gland regulation and also suggests the possibility of creating transgenic sheep using *JAK2* with improved milk fat and protein contents.

## Figures and Tables

**Figure 1 ijms-25-04027-f001:**
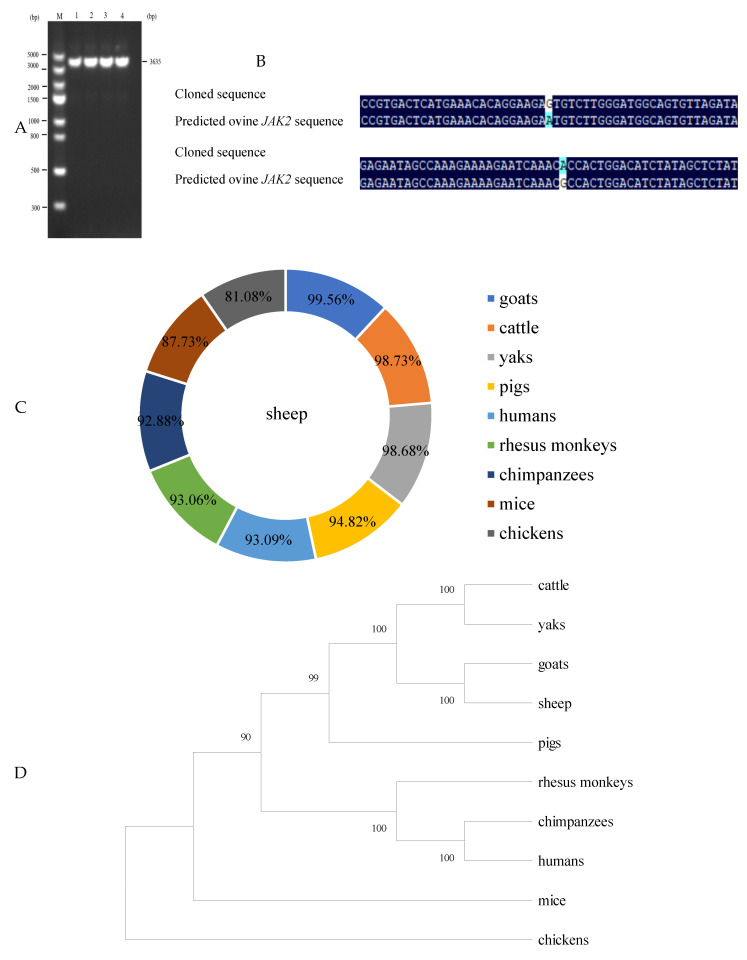
Cloning and nucleotide sequence analysis of ovine *JAK2*. (**A**) Agarose gel electrophoresis detection of the cloned coding sequence (CDS) region of ovine *JAK2*. M: DL5000 DNA marker; 1–4: ovine *JAK2* sequences. (**B**) Comparison of cloned sequences obtained in this study with predicted ovine *JAK2* sequences (XM_042243421.2). (**C**) Comparison of nucleotide homology between sequences obtained in this study and *JAK2* sequences from other animal species. (**D**) The neighbor-joining tree of sequences obtained in this study and *JAK2* sequences identified in other animal species. The cloned sequence in this study is shown in a box and *JAK2* sequences from other animal species include goats (XM_01805202), cattle (XM_003586385.6), yaks (XM_005891363.2), pigs (NM_214113.1), humans (NM_001322194.2), rhesus monkeys (NM_001265901.2), chimpanzees (XM_001139368.6), mice (NM_001048177.3), and chickens (NM_001030538.3).

**Figure 2 ijms-25-04027-f002:**
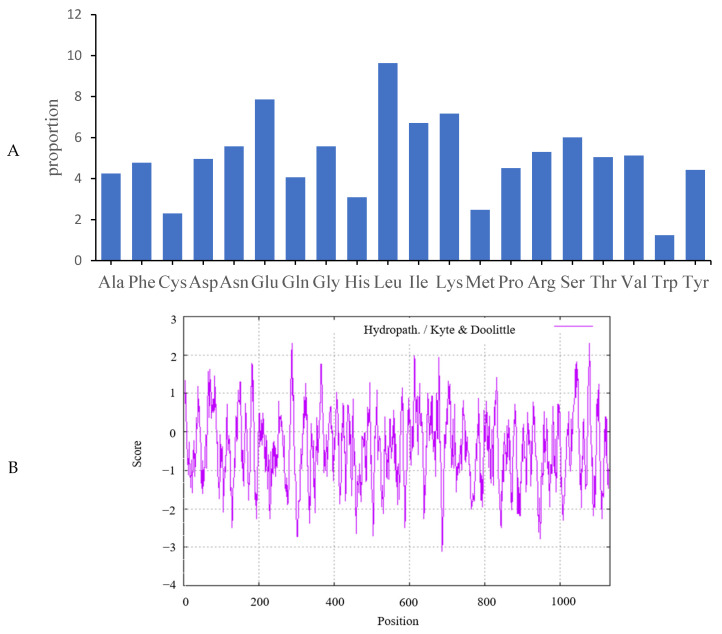
The characteristic analysis of the ovine JAK2 protein. (**A**) Amino acid composition analysis of the ovine JAK2 protein. (**B**) Hydrophilic and hydrophobic cluster analysis of the ovine JAK2 protein.

**Figure 3 ijms-25-04027-f003:**
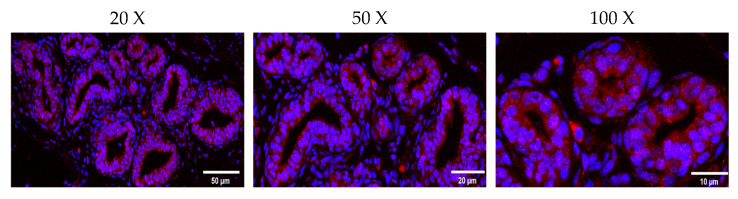
Immunofluorescence staining results of the JAK2 protein in ovine mammary gland tissue.

**Figure 4 ijms-25-04027-f004:**
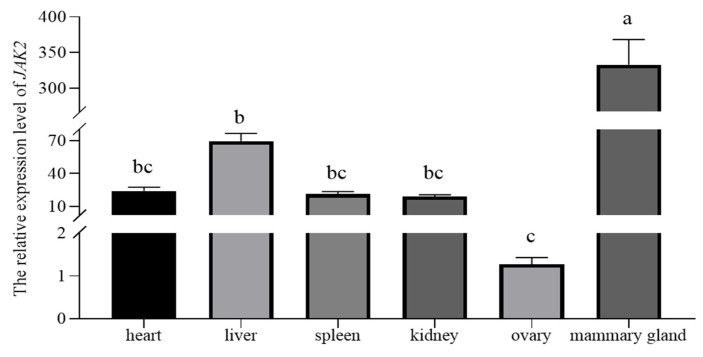
The relative expression level of *JAK2* in six ovine tissues. Different lowercase letters above the bars represent significant differences (*p* < 0.05).

**Figure 5 ijms-25-04027-f005:**
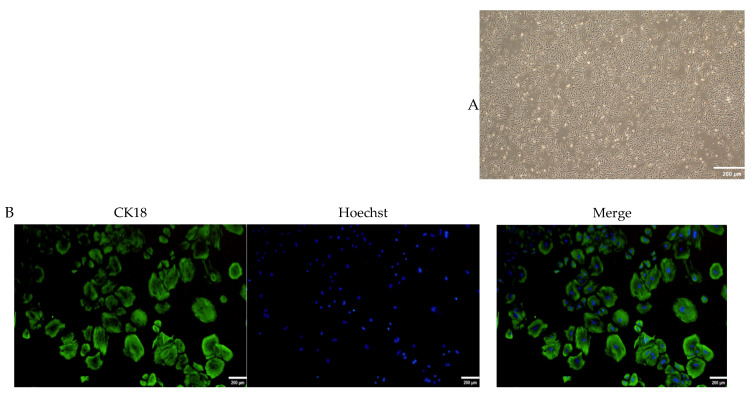
Culturing and identification of ovine mammary epithelial cells (OMECs). (**A**) The cultured OMECs. (**B**) Immunofluorescent identification of OMECs. Scale bar, 200 µm.

**Figure 6 ijms-25-04027-f006:**
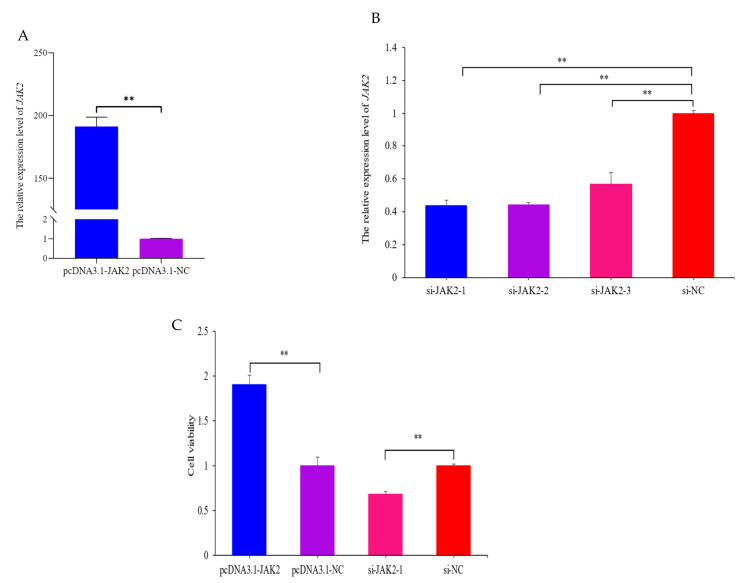
Effect of *JAK2* on viability of ovine mammary epithelial cells (OMECs). (**A**,**B**) Relative expression levels of *JAK2* when pcDNA3.1-*JAK2* and three groups of small interfered *JAK2* were transfected into OMECs. (**C**) Effect *JAK2* on the viability of OMECs, detected using a CCK-8 assay. ** *p* < 0.01.

**Figure 7 ijms-25-04027-f007:**
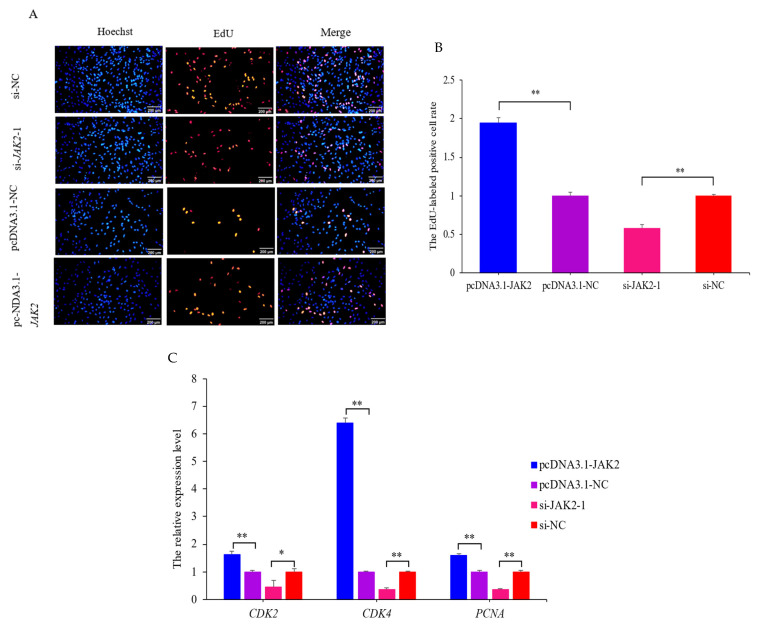
Effect of *JAK2* on the proliferation of ovine mammary epithelial cells (OMECs). (**A**) Effect of *JAK2* on the number of proliferated OMECs, detected using an EdU assay. Scale bar, 200 µm. (**B**) The percentage of EdU-labeled positive OMECs was calculated using ImageJ 1.54d software. (**C**) The expression levels of the cell proliferation marker genes *CDK2*, *CDK4*, and *PCNA* when pcDNA3.1-*JAK2* and si-*JAK2-*1 were transfected into OMECs. * *p* < 0.05 and ** *p* < 0.01.

**Figure 8 ijms-25-04027-f008:**
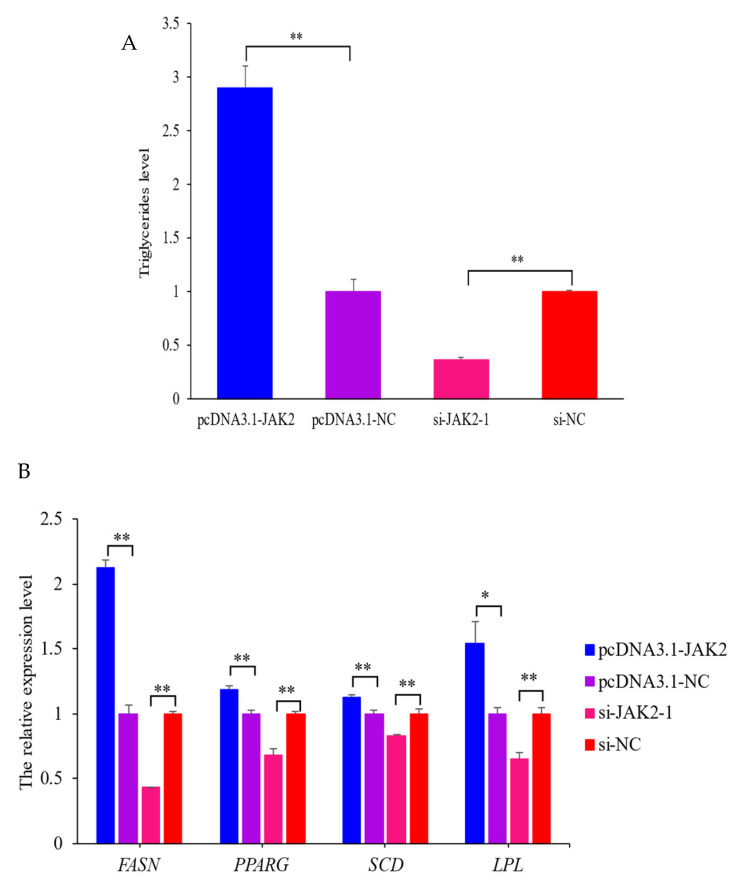
The triglyceride contents (**A**) and the expression levels of four milk fat synthesis marker genes (**B**) when pcDNA3.1-*JAK2* or si-*JAK2*-1 were transfected into ovine mammary epithelial cells (OMECs). * *p* < 0.05 and ** *p* < 0.01.

**Figure 9 ijms-25-04027-f009:**
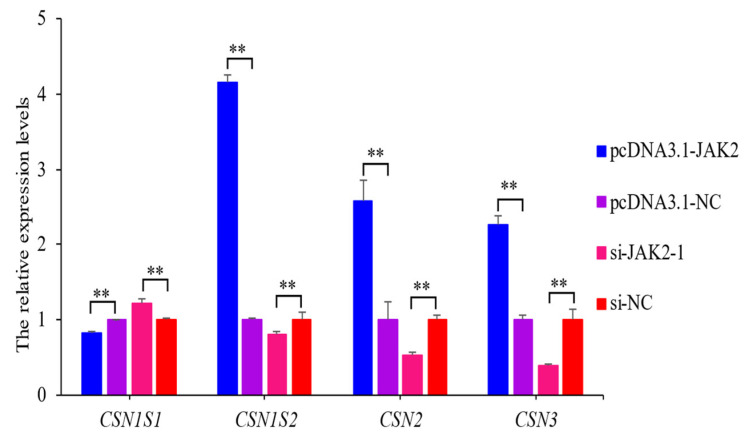
The expression levels of four casein genes when pcDNA3.1-*JAK2* or si-*JAK2*-1 were transfected into ovine mammary epithelial cells (OMECs). ** *p* < 0.01.

**Table 1 ijms-25-04027-t001:** Primer sequence information used for RT–qPCR.

Name	Forward (5′→3′)	Reverse (5′→3′)	Amplicon Size (bp)
*JAK2*	CTTGGGAAAGCTGAGGAGGA	CGATAGGTTCTGTTGCTGCC	260
*CDK2*	AGAAGTGGTGGCGCTTAAAA	TCTTGAGATCCTGGTGCAGA	185
*CDK4*	GCTTTTGAGCATCCCAATGT	AGGTCTTGGTCCACATGCTC	107
*PCNA*	GAACCTCACCAGCATGTCCAA	TTCACCAGAAGGCATCTTTACT	220
*FASN*	GGGCTCCACCACCGTGTTCCA	GCTCTGCTGGGCCTGCAGCTG	226
*SCD*	CGACGTGGCTTTTTCTTCTC	GATGAAGCACAACAGCAGGA	165
*PPARG*	CCGCTGACCAAAGCAAAG	GGAGCGAAACTGACACCC	189
*LPL*	AGGACACTTGCCACCTCATTC	TTGGAGTCTGGTTCCCTCTTGTA	169
*CSN3*	CAACAGAGACCAGTTGCACTAA	ACTTGGCAGGCACAGCATTT	130
*CSN2*	TCCCACAAAACATCCTGCCT	GGGAAGGGCATTTCCTTGTG	127
*CSN1S1*	AGCACCAAGGACTCTCTCCA	TGACGAACTGCTTCCAGCTT	182
*CSN1S2*	AATCTGCTGAAGTTGCCCCA	TGGGCCTTGATACAGATACTGG	131
*β-actin*	AGCCTTCCTTCCTGGGCATGGA	GGACAGCACCGTGTTGGCGTAGA	113

**Table 2 ijms-25-04027-t002:** The siRNA sequences of *JAK2*.

Name	Forward (5′–3′)	Reverse (5′–3′)
si-*JAK2*-1	GCAAAUAGAUCCAGUCCUATT	UAGGACUGGAUCUAUUUGCTT
si-*JAK2*-2	GCAAAUCAAGAAGGCGCAATT	UUGCGCCUUCUUGAUUUGCTT
si-*JAK2*-3	CCAGCUUUCUCACAAACAUTT	AUGUUUGUGAGAAAGCUGGTT
si-NC	UUCUCCGAACGUGUCACGUdTdT	ACGUGACACGUUCGGAGAAdTdT

## Data Availability

All data presented in this study are available within the article.

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
