# Peer review of "Molecular Characteristics of JAK2 and Its Effect on the Milk Fat and Casein Synthesis of Ovine Mammary Epithelial Cells"

_ijms, 2024, doi:10.3390/ijms25074027_

Round 1

Reviewer 1 Report

Comments and Suggestions for Authors

The article by Yuan Liu and coworkers, entitled “Molecular characteristics of JAK2 and its effect on milk fat and casein synthesis of ovine mammary epithelial cells” describes an in vitro study proving the direct relationship between the expression of Janus kinase 2 (JAK2) and the proliferative activity as well as functional differentiation of ovine mammary epithelial cells (OMECs). The study was conducted on primary ovine epithelial cells isolated from the mammary glands of ewes, and includes a wide range of molecular and cell biology techniques, such as: cloning the coding sequences of JAK2 gene and characteristic of the amplified of the specific CDS region; experiments with ovine mammary epithelial cells overexpressing JAK2 or with JAK2 gene knock down; analysis of expression of genes encoding milk proteins or enzymes and transcription factors involved in milk fat synthesis; classic cell based assays (proliferation assay, cell viability assay). All performed experiments proved the direct relationship between the JAK2 expression and increased proliferative activity of MECs, increased expression of major milk proteins  (caseins: alphaS2, beta and kappa), as well as increased expression of genes encoding fatty acid synthase, stearoyl-CoA desaturase, lipoprotein lipase and PPARgamma. These finding are not novel, as the function of JAK/STAT5 signaling pathway in the mammary gland development is well documented. The fact that prolactin induces JAK-STAT pathway leading to transcription of genes that induce alveolar proliferation and differentiation was extensively studied in the 1990’s, first using animal models of rodent mammary glands, then, since the early years of the XXI century, the findings have been confirmed in the mammary glands of cows and goats. The real novelty of this study is the direct confirmation of JAK2 functions specifically in the ovine mammary epithelial cells, although researchers could have expected such results based on the studies performed on other diary animals. The Authors also did not mention a study by Amy M. Paten and coworkers published in 2015, describing results of the RNA-seq to identify differences in gene expression in the ovine mammary gland between late pregnancy and lactation (Paten AM, Duncan EJ, Pain SJ, Peterson SW, Kenyon PR, Blair HT, Dearden PK. Functional development of the adult ovine mammary gland-insights from gene expression profiling. BMC Genomics. 2015;16:748. doi: 10.1186/s12864-015-1947-9). In this study the research group showed that JAK-STAT and PPAR pathways, were differently regulated, indicating key roles for these pathways in functional development of the ovine mammary gland.

The Authors have made a substantial effort and plenty of laboratory work to confirm the important role of JAK2-mediated pathway in the ovine mammary gland development. For this reason the study is worth publishing. However, the current version of the manuscript is badly written and needs extensive editing before potential publication. Detailed remarks are listed below:

1.      There are plenty of grammar and linguistic errors in the text. The Authors have to correct English grammar, and rephrase the majority of sentences for better clarity.

2.      The Authors should change the speculative phrases, such as “these suggest”, ”these findings indicate” when describing the results that simply confirm the well documented JAK2 functions in the mammary glands of other species.

3.      The Authors declare that “The cloned sequences would encode polypeptides of 1,132 amino acids in length and it  shared the highest homology with the JAK2 sequences originated from goats, cattle and  yaks” (lines: 386-387). The in silico analysis of JAK2 amino acid sequence homology  should be presented in the results.

4.      Lines: 209-210: when describing the morphology of OMECs, the authors should use the expression “cobblestone morphology”, not “paving stone-like OMECs”

5.      Lines: 401-405: immunofluorescence staining confirmed the expression of JAK2 protein mainly in the luminal mammary epithelial cells. The Authors should properly define the localization, instead of describing it as: “MECs around the mammary alveolus, which is an important place for milk synthesis”. Mammary alveoli are the basic secretory units in the functionally active mammary gland, and the milk components are synthesized only by the luminal MECs.

6.      The Authors should provide more detailed information about the age of ewes, parity, stage of lactation at which samples of the mammary gland tissue were collected.

7.      Were the antibodies used for JAK2 detection specific for sheep? It seems that if the JAK2 sequence is only predicted, then the antibodies were not specifically designed to detect ovine JAK2 protein. In that case was the cross reactivity  indicated by the producer?

8.      Line 519: “Immunofluorescence technology was employed to determine the localization of JAK2 …” rephrase the sentence accordingly:  “The localization of JAK2 in the ovine mammary gland was detected by immunofluorescence staining”

9.      What do Authors mean by the last sentence of conclusions (lines: 591-592): “ … also further lay a theoretical foundation for improving milk fat and protein contents by regulating the expression of JAK2 in sheep.” Does this suggest the possible use of JAK2 gene to create transgenic animals?  I think that this conclusion is too far-reaching.  

Comments on the Quality of English Language

There are plenty of grammar and linguistic errors in the text. The Authors have to correct English grammar, and rephrase the majority of sentences for better clarity.

Author Response

Dear Editor

Thank you for the opportunity to address the concerns of the editor and the reviewers. We have answered each of their questions below and modified the manuscript and diagram using ‘trach changes’ in WORD.

Editor:

As we were conducting our initial checks, we noticed that the self-citation rate in your work is slightly too high for the journal standards. In light of this, we politely request that you consider minimizing the self-citation rate to the best of your abilities, preferably by adding several recent (from the past 5 years) references from other specialists in the field. The ratio of self citation should be less than 15%. In the academic world, citing a diverse range of sources is of paramount importance, and we take great pride in ensuring that our authors uphold the highest standards of academic integrity.

AU: We have reduced the ratio of self-citation as much as possible by adding references 10, 27, 28 and 36.

Reviewer #1:

The article by Yuan Liu and coworkers, entitled “Molecular characteristics of JAK2 and its effect on milk fat and casein synthesis of ovine mammary epithelial cells” describes an in vitro study proving the direct relationship between the expression of Janus kinase 2 (JAK2) and the proliferative activity as well as functional differentiation of ovine mammary epithelial cells (OMECs). The study was conducted on primary ovine epithelial cells isolated from the mammary glands of ewes, and includes a wide range of molecular and cell biology techniques, such as: cloning the coding sequences of JAK2 gene and characteristic of the amplified of the specific CDS region; experiments with ovine mammary epithelial cells overexpressing JAK2 or with JAK2 gene knock down; analysis of expression of genes encoding milk proteins or enzymes and transcription factors involved in milk fat synthesis; classic cell based assays (proliferation assay, cell viability assay). All performed experiments proved the direct relationship between the JAK2 expression and increased proliferative activity of MECs, increased expression of major milk proteins (caseins: alphaS2, beta and kappa), as well as increased expression of genes encoding fatty acid synthase, stearoyl-CoA desaturase, lipoprotein lipase and PPARgamma. These finding are not novel, as the function of JAK/STAT5 signaling pathway in the mammary gland development is well documented. The fact that prolactin induces JAK-STAT pathway leading to transcription of genes that induce alveolar proliferation and differentiation was extensively studied in the 1990’s, first using animal models of rodent mammary glands, then, since the early years of the XXI century, the findings have been confirmed in the mammary glands of cows and goats. The real novelty of this study is the direct confirmation of JAK2 functions specifically in the ovine mammary epithelial cells, although researchers could have expected such results based on the studies performed on other diary animals. The Authors also did not mention a study by Amy M. Paten and coworkers published in 2015, describing results of the RNA-seq to identify differences in gene expression in the ovine mammary gland between late pregnancy and lactation (Paten AM, Duncan EJ, Pain SJ, Peterson SW, Kenyon PR, Blair HT, Dearden PK. Functional development of the adult ovine mammary gland-insights from gene expression profiling. BMC Genomics. 2015;16:748. doi: 10.1186/s12864-015-1947-9). In this study the research group showed that JAK-STAT and PPAR pathways, were differently regulated, indicating key roles for these pathways in functional development of the ovine mammary gland.

AU: We have mentioned a study by Amy M. Paten and coworkers published in 2015 in lines 68-71.

The Authors have made a substantial effort and plenty of laboratory work to confirm the important role of JAK2-mediated pathway in the ovine mammary gland development. For this reason the study is worth publishing. However, the current version of the manuscript is badly written and needs extensive editing before potential publication. Detailed remarks are listed below:

  1. There are plenty of grammar and linguistic errors in the text. The Authors have to correct English grammar, and rephrase the majority of sentences for better clarity.

AU: We have used an English editing service suggested by the IJMS journal to correct English grammar and linguistic errors in the text and rephrased some sentences for better clarity.

  1. 2. The Authors should change the speculative phrases, such as “these suggest”, ”these findings indicate” when describing the results that simply confirm the well documented JAK2 functions in the mammary glands of other species.

AU: We have changed the speculative phrases in lines 25, 33, 114, 176, 212, 234, 264, 301-302, 343, 376.

  1. 3. The Authors declare that “The cloned sequences would encode polypeptides of 1,132 amino acids in length and it shared the highest homology with the JAK2 sequences originated from goats, cattle and yaks” (lines: 386-387). The in silico analysis of JAK2 amino acid sequence homology should be presented in the results.

AU: We analyzed the similarities of the nucleotide sequences obtained in the study with JAK2 gene sequences from other species. Meanwhile, we constructed a phylogenetic tree using nucleotide sequences from the study and JAK2 gene sequences from other species. In this context, we have presented the in silico analysis results of JAK2 gene nucleotide sequence homology in the Results section in lines 107-114, 132-156.

  1. 4. Lines: 209-210: when describing the morphology of OMECs, the authors should use the expression “cobblestone morphology”, not “paving stone-like OMECs”

AU: We have replaced the wordings in line 231.

  1. 5. Lines: 401-405: immunofluorescence staining confirmed the expression of JAK2 protein mainly in the luminal mammary epithelial cells. The Authors should properly define the localization, instead of describing it as: “MECs around the mammary alveolus, which is an important place for milk synthesis”. Mammary alveoli are the basic secretory units in the functionally active mammary gland, and the milk components are synthesized only by the luminal MECs.

AU: We have clearly defined the localization in lines 413-415.

  1. 6. The Authors should provide more detailed information about the age of ewes, parity, stage of lactation at which samples of the mammary gland tissue were collected.

AU: We have provided more detailed information about the age of ewes, parity, stage of lactation at which samples of the mammary gland tissue were collected in lines 506-508.

  1. 7. Were the antibodies used for JAK2 detection specific for sheep? It seems that if the JAK2 sequence is only predicted, then the antibodies were not specifically designed to detect ovine JAK2 protein. In that case was the cross reactivity indicated by the producer?

AU: As the reviewer mentioned, the JAK2 protein used in this study was indeed not specifically designed for ovine JAK2 protein detection. We have contacted the manufacturer and confirmed that the homology between the antibody used in the study and sheep sequences is 100% and the cross reactivity was indicated by the producer.

  1. 8. Line 519: “Immunofluorescence technology was employed to determine the localization of JAK2 …” rephrase the sentence accordingly: “The localization of JAK2 in the ovine mammary gland was detected by immunofluorescence staining”.

AU: We have rephrased the sentence in lines 554-555.

  1. 9. What do Authors mean by the last sentence of conclusions (lines: 591-592): “ … also further lay a theoretical foundation for improving milk fat and protein contents by regulating the expression of JAK2 in sheep.” Does this suggest the possible use of JAK2 gene to create transgenic animals? I think that this conclusion is too far-reaching.

AU: We have rewritten the sentences in lines 630-632.

Reviewer #2:

The manuscript descrive la caratterizzazione del gene JAK2 in Hu sheep. Gli autori hanno sequenziato l’RNA codificante JAK2 e hanno valutato il profilo di espressione in diversi tessuti, inoltre hanno valutato l’effetto di questo gene sull’espressione di geni protein coding e lipogenic in OMECs. The topic is of interest to the scientific community, an ddeserves to be published.

AU: Thank you very much!

Some minor issues,

  1. cows with GG genotype” please rephrase, genotype has to be indicated correctly?

AU: We have rephrased the sentences in lines 86-88.

  1. 2.2. Ovine JAK2 protein characteristics analysis” please include the word “predicted”.

AU: We have added the wording “predicted” in line 167.

  1. please check Figure 5 A.

AU: We have replaced Figure 5A with a clear picture and revised the format of the Figure5A in lines 237-249.

Reviewer 2 Report

Comments and Suggestions for Authors

Revision of Manuscript entitled “Molecular characteristics of JAK2 and its effect on milk fat and 2

casein synthesis of ovine mammary epithelial cells”

The manuscript descrive la caratterizzazione del gene JAK2 in Hu sheep. Gli autori hanno sequenziato l’RNA codificante JAK2 e hanno valutato il profilo di espressione in diversi tessuti, inoltre hanno valutato l’effetto di questo gene sull’espressione di geni protein coding e lipogenic in OMECs. The topic is of interest to the scientific community, an ddeserves to be published.

Some minor issues,

Line 76: “cows with GG genotype” please rephrase, genotype has to be indicated correctly

Line 149: “2.2. Ovine JAK2 protein characteristics analysis” please include the word “predicted”

Line 236: please check Figure 5 A.

Author Response

(The authors gave the same response as above.)

Round 2

Reviewer 1 Report

Comments and Suggestions for Authors

The manuscript entitled “Molecular characteristics of JAK2 and its effect on milk fat and casein synthesis of ovine mammary epithelial cells” (Manuscript ID: ijms-2908578) has been largely improved. The revised version, corrected by an  English editing service is easy and pleasant to read. This improved the general impression of the presented study, and emphasized the value of results obtained.

The Authors implemented all suggested changes.

In my opinion the new version of the manuscript should be presented to the general public.

Well done dear Authors!

Author Response

The manuscript entitled “Molecular characteristics of JAK2 and its effect on milk fat and casein synthesis of ovine mammary epithelial cells” (Manuscript ID: ijms-2908578) has been largely improved. The revised version, corrected by an  English editing service is easy and pleasant to read. This improved the general impression of the presented study, and emphasized the value of results obtained.

The Authors implemented all suggested changes.

In my opinion the new version of the manuscript should be presented to the general public.

Well done dear Authors!

AU: Thank you very much!